# Experimental Determination of a Chiral Ternary Solubility Diagram and Its Interpretation in Gas Antisolvent Fractionation

**DOI:** 10.3390/molecules28052115

**Published:** 2023-02-24

**Authors:** Márton Kőrösi, Csaba Varga, Péter Tóth, Noémi Buczkó, Erzsébet Varga, Edit Székely

**Affiliations:** 1Department of Chemical and Environmental Process Engineering, Faculty of Chemical Technology and Biotechnology, Budapest University of Technology and Economics, Műegyetem rkp. 3., H-1111 Budapest, Hungary; 2Cyclolab Cyclodextrin Research and Development Laboratory Ltd., 7. Illatos út, H-1097 Budapest, Hungary

**Keywords:** ternary solubility plot, eutonic composition, enantiomeric enrichment, self-disproportionation of enantiomers

## Abstract

Although crystallization has been widely applied for the enantiomeric enrichment of non-racemates both in research and in industrial applications, the physical–chemical background of chiral crystallizations is not as frequently discussed. A guide for the experimental determination of such phase equilibrium information is lacking. In the current paper, the experimental investigation of chiral melting phase equilibria, chiral solubility phase diagrams and their application in atmospheric and supercritical carbon dioxide-assisted enantiomeric enrichment is described and compared. Benzylammonium mandelate is a racemic compound; it shows eutectic behavior when molten. A similar eutonic composition was observed in its methanol phase diagram at 1 °C. The influence of the ternary solubility plot could be unequivocally discovered in atmospheric recrystallization experiments, which proved that the crystalline solid phase and the liquid phase were in an equilibrium. The interpretation of the results obtained at 20 MPa and 40 °C, using the methanol–carbon dioxide mixture as a pseudo-component, was more challenging. Although the eutonic composition was found to be the limiting enantiomeric excess value in this purification process as well, the high-pressure gas antisolvent fractionation results were only clearly thermodynamically controlled in certain concentration ranges.

## 1. Introduction

Although chirality is a widely known phenomenon among organic chemists—it has been studied since the 19th century [1]—its manifestations still fascinate a large number of researchers from various chemical fields. New publications on exploiting chirality in pharmaceuticals, polymers and catalysis appear in large numbers [2,3,4,5], but traditionally known and applied techniques may also hold a few details still to be explored. Enantiomers, the mirror-image forms of chiral molecules, often only differ in the sterical arrangement of four different groups bound to a carbon atom (central chirality). Enantiomeric forms of the same molecule often exhibit dramatically different biological effects, which is a key issue in modern pharmaceutical and agrochemical industries, among others. A straightforward separation method of enantiomers is fractioned crystallization, which is often governed by thermodynamic effects [6,7,8]. Such effects stem from the interaction and associate-formation capabilities of chiral molecules with their own species.

Racemic compounds, a group of chiral chemicals, are characteristic of stronger interactions between two molecules of different enantiomers than those between identical ones. The difference between the stability of heterochiral and homochiral associates accounts for the melting and dissolution behavior of racemic compounds [6,7]. Their chiral melting phase diagram shows eutectic behavior and the melting temperature of the racemate is usually higher than that of the pure enantiomer. As Kellogg and Leeman discuss in their exhaustive review on the topic of chirality and enantioseparation [7], the solubility of enantiomeric mixtures is strongly connected to their melting behavior. This is particularly important when enantiomeric mixtures are purified, aiming for a pure enantiomer, using multistep fractioned recrystallization. Due to the uneven interactions between identical enantiomers and antipodes, the enantiomeric composition of the product of fractioned recrystallization is often different compared with the starting material. The phenomenon occurs in a variety of physical–chemical fractionation procedures and is called self-disproportionation of enantiomers (SDE) [9]. Although SDE has already been exploited in various processes in the production of high-purity enantiomers, it is also worth noting that it may distort scientific results. For this reason, tests regarding SDE have been proposed [10,11]. As a rule of thumb, it can be assumed that the higher the melting temperature, the lower the solubility. The eutectic composition also has a special role. In thermodynamically controlled systems, it may not be crossed by fractioned recrystallization alone. A eutectic mixture of enantiomers can also be expected to have the highest solubility. The phase equilibrium conditions of crystallizing enantiomeric mixtures in solvents can be discussed using a ternary diagram. A schematized ternary solubility phase diagram [7] compared with a melting phase diagram [6] is presented in Figure 1. The expected outcome of enantiomeric enrichment is shown in the form of product ‘*ee*’ in the initial ‘*ee*’ diagrams. This set of three diagrams shows idealized cases. An extended introduction of the ternary solubility diagram has been presented by Kellogg and Leeman. [7] Two nodes of the triangle represent one enantiomer or the other, whilst the third node refers to the solvent. As usual, the sides refer to binary mixtures. In the case of racemic compounds, the biphasic (solid–liquid) area of the diagram is located under the red, solid curve. It is divided into three zones (exploiting the symmetry of the diagram). They are separated by solid green lines and they differ in the phase determining the equilibrium compositions. Point ‘a’ represents a solid enantiomeric mixture. The line between ‘a’ and the solvent node shows the possible total compositions of solutions prepared from ‘a’. The dashed lines show the equilibrium compositions in two different heterogeneous regions of the diagram. If the total composition is set to ‘b’, the mixture point lies in zone ‘2’ of the diagram. In this zone, the composition of the mother liquor is constant (eutonic) and it determines the composition of the crystalline phase as well. In region ‘1’, the crystalline phase (‘f’) becomes racemic, whilst the mother liquor (‘g’) contains a non-racemic mixture of enantiomers. In region ‘3’, an enantiomerically pure crystalline phase may be expected. It must be noted that, by altering the concentration of the solute, the mixture point may fall in different regions of the diagram, thus allowing for different enantiomeric compositions in the products.

Experimental investigations of such ternary solubility equilibria are less frequent than investigations on chiral separation. However, conducting multiple sets of recrystallization experiments can reveal the most important features of the diagram. In a set, the total concentration of the solute should be identical, whilst its overall enantiomeric composition should range from close to racemic to almost enantiomeric. The multiple sets should differ in the total mixture composition. The mixture points (i.e., the total composition of the solutions prior to their crystallization) lie on horizontal straight lines on the ternary equilibrium diagram. By investigating the composition of the crystalline and dissolved products, one can identify the zones of the ternary phase diagram. The expectable results of such crystallization sets, starting with more and more dilute solutions, are shown on the right side of Figure 1. If the crystalline product of an individual recrystallization is (nearly) racemic with a scalemic mixture dissolved in the mother liquor, the points belong in the first zone. A constant enantiomeric composition in the mother liquor (close to the eutectic ‘*ee*’) is the indicator of the second zone. A scalemic precipitate may be expected. In the third zone, the enantiomeric excess in the mother liquor decreases, but a (nearly) enantiomeric crystalline phase may be obtained.

In addition to the contribution to the fundamental aspects of chiral resolution, such ternary equilibrium diagrams may aid the design of multistep recrystallization processes.

Whilst recrystallization-based enantiomeric enrichment procedures have traditionally been carried out in solvents at an ambient pressure, the application of high-pressure precipitation processes for enantioseparation has also been investigated [12,13,14,15,16,17,18,19]. In these earlier studies, carbon dioxide was utilized as an antisolvent in the precipitation of polar organic components from their solutions. Among the multiple variations in carbon dioxide antisolvent techniques [20], the simplest is batch-type gas antisolvent precipitation. It can be conducted in a simple autoclave; thus, it is well-suited for laboratory and fundamental studies. The most important step of the procedure is mixing the original organic solvent with carbon dioxide. The solubility of the solute (provided that it is sufficiently polar) decreases in the forming mixture, resulting in its precipitation. Solid formation takes place in a matter of minutes. During the carbon dioxide extraction of the organic solvent, the composition of the solvent mixture shifts. With an increase in the proportion of carbon dioxide, the quantity of the solid may increase. The operational parameters—including the concentration of the organic solvent, pressure and temperature—may be set to keep a desired fraction of the solute from precipitating. If a considerable fraction of the target material is extracted, the process is called gas antisolvent fractionation (GASF). The ability to control the morphology of the solid particles is widely considered to be the main advantage of carbon dioxide antisolvent techniques. The tunable physico-chemical properties of carbon dioxide and the possibility of reducing the necessary amount of organic solvents are common features of several innovative and applied processes such as supercritical fluid extraction [21].

In earlier studies aiming for the enantiomeric enrichment of mandelic acid derivatives, GASF proved to be feasible for partial recrystallization [12,22,23,24]. A somewhat counter-intuitive, clear correlation was found between the chiral melting phase diagrams and the enantiomeric enrichment curves of the chosen model molecules. In our current study, we aimed to investigate if and how the atmospheric ternary solubility diagram affected the outcome of the rapid crystallization process.

Mandelic acid is a well-known and simple model compound in the investigation of chiral separation processes [25,26,27,28,29,30]. Whilst it is an ideal candidate for phenomenological research, it also bears practical relevance as an intermediate [31,32] in the pharmaceutical industry and is also used as an agent against infections or dermal symptoms [33,34] by itself. Racemic mandelic acid has been successfully resolved using 1-phenylethylamine, even by high-pressure precipitation methods [14]. Its melting equilibrium is also known. As a racemic compound, it shows a eutectic composition of around 40% *ee* [35]. It is common practice among organic chemists working on chiral resolution to shift the eutectic composition of racemic compounds by adding an achiral reactant. A well-known example for the phenomenon is combining ibuprofen with sodium [6]. Adding benzylamine to mandelic acid could have the same effect. As discussed later, the eutectic composition of benzylammonium mandelate lies around 94–98% *ee*. This means that the original eutectic composition of mandelic acid may be circumvented by the multistep fractioned recrystallization of its salt with a simple and inexpensive achiral agent. The described study involved the partial recrystallization of several samples of the scalemic salts of mandelic acid and benzylamine. The samples were prepared in an equimolar reaction of the compounds and only differed in their enantiomeric composition; they are referred to as ‘salt’ later on.

## 2. Results and Discussion

### 2.1. Chiral Melting Phase Diagram

The temperature-to-composition phase diagram of benzylammonium mandelate is shown in Figure 2. Triangles mark the temperatures that were associated with the eutectic melting phenomenon. The onset temperatures of the higher temperature and larger melting peaks are plotted as circles. The solid line shows the values that were calculated using the Schröder–van Laar and Prigogine–Defay equations [36] based on the measured melting temperatures and fusion enthalpies of the pure enantiomer (166.6 °C; 38,000 J/mol) and those of the racemate (180.9 °C; 40,400 J/mol). Both the measured data and the prediction confirmed that benzylammonium mandelate was a racemic compound and had a eutectic composition of around 94–98% *ee*. The minimum of the predicted liquidus curve appeared at 96.9% *ee*. However, the measured melting temperatures only loosely fitted the prediction. High-temperature polymorphism may be an explanation for the phenomenon.

### 2.2. Atmospheric Recrystallization Experiments

As demonstrated in Figure 1, a series of recrystallization experiments could be used to discover the zones and boundaries present in the ternary solubility plots. Recrystallization series were conducted at different total concentrations of the scalemic benzylammonium mandelate salt. Within such series, the initial enantiomeric excess was systematically changed in order to identify the zones of the ternary solubility diagram. The markers on the individual purification plots were assigned to the zones of the ternary diagram in the order and shape in which they are shown in Figure 1. 

The equimolar salts of scalemic mandelic acid and benzylamine were used to prepare solutions of different concentrations. The effect of the initial enantiomeric excess on the chiral composition in the products was studied in detail. The results obtained at various total concentrations are shown in Figure 3. 

The triangular markers show the experiments where a racemic solid was expected (zone 1). The circles correspond with compositions where one may expect a eutonic mother liquor and the diamonds show the results with an enantiomerically pure solid phase. Naturally, the exact enantiomeric excess values were loaded with an error resulting from the sample processing and chiral analytics. Whilst it was possible to perform crystallization in the third region of the ternary solubility plot in the concentration range of 124.9 mg/mL to 58.5 mg/mL, no crystalline product was formed when the overall initial concentration of the salt was decreased to 29.38 mg/mL. At higher than approximately 90% enantiomeric excess values, the mixture point apparently fell in the homogeneous region of the ternary solubility diagram.

By assigning the composition ranges in each *ee*_1_–*ee*_0_ diagram to a region of the ternary solubility plot, it was possible to approximate the ternary diagram itself. The close vicinity of the eutectic composition of benzylammonium mandelate to the total enantiomeric purity did not allow the mathematical description of the diagram. Hence, the region boundaries were illustrated in the form of Bézier curves. The mass of the solute in the mother liquor (and, hence, its mole fraction in the methanol solution) was calculated from that of the crystalline product after filtration. This method (despite being practical in vial-scale experiments at all of the applied enantiomeric compositions) probably overestimated the concentration in the equilibrium solutions. As atmospheric experiments were conducted at a 10 mL scale range, the mass balance of the experiments may have been loaded with an error. Our estimate of the ternary solubility plot is shown in Figure 4, which contains the measured composition of the mother liquors separated in each experiment. The diagram is magnified to present the vicinity of the solvent’s node.

The boundary lines of the diagram’s sections were set in the following order. The points showing an equilibrium between a racemic solid phase and a scalemic mother liquor (triangular markers (‘a’)) formed a clearly followable tendency; hence, a Bézier curve was used as their purely empirical description. According to the chiral melting phase equilibrium of the salt, the eutonic composition was expected to be around 97% *ee*. All the solutions that contained the salt with a eutectic *ee* could be represented as the straight line connecting the MeOH corner and the point for this enantiomeric excess value on the axis of the solid materials. The points marked by empty circles (‘b’) show individual experiments where a eutonic mother liquor was expected. Due to the mass balance error and the large magnification of the ternary diagram, these points were scattered in a seemingly wide range. Their average was plotted as a solid circle (‘c’). The solubility of the pure enantiomer was plotted as an X (‘f’). The composition of the mother liquors recovered in experiments with (nearly) enantiopure crystals were plotted as empty diamonds (‘d’). Their average is shown as a solid diamond marker (‘e’). It must be noted that the solubility of the pure enantiomer had to be lower compared with that of a eutectic enantiomeric mixture. Due to this, and also due to the fact that the solid diamond marker lays on the line of the eutectic composition, we decided to use this point as an approximation of the eutonic composition when setting the boundaries of the phase diagram. Connecting it to the solubility point of a pure enantiomer and to the *R*-corner of the diagram, the boundaries of the third region were formed. Connecting the eutonic composition to the point of a solid racemate (not shown in the diagram due to its magnification), the second region became visible; the first region was enclosed by the curve drawn on the empty circular markers and the line drawn previously.

Such triangular solubility diagrams can be used not only to understand and explain the results of enantiomeric enrichment experiments, but also (if already constructed) to determine a multistep recrystallization path in an industrial environment. We believe that the diagram that we constructed can serve the purpose of identifying the regions and predicting the expectable course of recrystallization experiments. However, it is clearly visible that the accurate description of such equilibrium processes would necessitate larger-scale preparative experiments with an accurate determination of the composition of the mother liquor.

### 2.3. Enantiomeric Enrichment by GASF

Gas antisolvent fractionation, a high-pressure, rapid crystallization method, has been proven to be effective in enantiomeric enrichment. Antisolvent fractionation procedures, despite being high-pressure processes, may find their application in crystallization tasks where the product has a high added value. They allow for a considerably lower processing time compared with atmospheric crystallization, whilst also offering a micronized product with a low residual solvent content and an often controllable morphology.

The correlation between high-pressure enantiomeric enrichment experiments and atmospheric, and even high-pressure, melting phase diagrams has been demonstrated [23,24,37,38]. This correlation may mean that very similar solubility equilibria determine the composition of the products as in thermodynamically controlled atmospheric recrystallizations. We investigated whether the effects of the different zones could be discovered on the product *ee* and initial *ee* diagrams of high-pressure experiments.

Similar to atmospheric crystallization tests, high-pressure, quick partial recrystallizations were carried out at various concentrations of the scalemic salt. The initial enantiomeric excess values were varied. Although the concentration of the salt in the reactor was different in all three experimental series, that of methanol was kept constant at approximately 90 mg/mL. The pressure and temperature were 20 MPa and 35 °C, respectively, in every experiment. As in Figure 3, the Product *ee* against Initial *ee* diagrams were recorded at different total salt concentrations and are shown in Figure 5. From panel (a) to (c), the diagrams are arranged in the decreasing order of the overall concentration of the scalemic benzylammonium mandelate salt. The markers distinguish the expected regions of the ternary phase diagram wherever they could unequivocally be assigned to the measurement result. The markers shaped as an ‘X’ show that, based on the evaluation of the enantiomeric composition of the raffinates and extracts, the experiment could not be categorized into any of the regions of the ternary solubility diagram.

In panel (a), it can be seen that the effect of the thermodynamic control and the ternary plot was spectacular. The enantiomeric purities of the extracts were very similar and were close to the expected eutectic composition of the salt. Therefore, it could be assumed that the total composition of the mixtures in the reactor lays in region ‘2’ of the ternary plot. Interpreting the data in panels (b) and (c) was not as straightforward. In the diagram of panel (b) (the concentration of the salt in the reactor was 1.15 mg/mL), only a few experiments resulted in eutonic extracts. The measurements below a 50% initial *ee* could not be straightforwardly categorized. Whilst the non-racemic (and non-eutonic) extracts suggested that they belonged in region ‘1’, the raffinates were also non-racemic, which was only characteristic for region ‘2’. The same tendency could be observed when the total concentration of the salt was further decreased (panel c).

Although the results at high salt concentrations confirmed the applicability of the ternary dissolution phase diagram, the enantiomeric excess values obtained at lower salt concentrations could not be so easily connected to the phase diagram. As these experiments were generally conducted at lower than eutectic initial enantiomeric excesses of the salt, they could be expected to have a mixture point in either the first or second region of the ternary solubility plot. The first region would be identified by the constant (approximately 0%) *ee* in the raffinate whereas in the second one, a constant extract *ee* would be expectable. In the experiments shown as ‘unident.’ in the diagrams of Figure 5, the enantiomeric composition of both products shifted. Although a single ternary solubility plot is not enough to fully explain the phenomenon, imagining multiple ones may serve as an explanation.

During the extraction phase of antisolvent fractionation, the composition of the fluid phase changes. The organic solvent, serving as an entrainer, is washed out. In other words, the pseudo-component in the solvent corner of the diagram changes in a continuous fashion during the extraction phase. The solubility of the benzylammonium mandelate salt decreases as the extraction proceeds. As the eutectic/eutonic enantiomeric excess is not expected to change (to a large extent), the first region of the ternary solubility plot narrows down gradually. Thus, it is theoretically possible that the mixture point of an experiment falls in region ‘1’ in the equilibration phase, but transfers into region ‘2’ during the extraction phase. Proving this theory could explain the tendencies of the raffinate and extract compositions in the (b) and (c) panels of Figure 5.

### 2.4. Parameter Testing in GASF

As mentioned in the above paragraph, the dissolving capability of a high-pressure solvent mixture may significantly influence the solubility equilibrium. In the ternary solubility diagram schematized in Figure 1, the high-pressure solvent mixture could be imagined as a pseudo-component. When the ratio of the organic modifier to carbon dioxide was different, this pseudo-solvent slightly changed.

The recrystallization of racemic benzylammonium mandelate via GASF was conducted to set the optimal operational parameters for enantiomeric enrichment. We looked for a combination of pressure, temperature and organic solvent concentration to achieve a significant amount of extract to observe enantiomeric enrichment due to the self-disproportionation of enantiomers. In the studies on the effect of temperature and pressure on the yield of the crystalline raffinate and extract, the pressure was altered between 12 and 20 MPa and the temperature levels were chosen to be 35 and 45 °C. Neither of these operational parameters had a significant effect on the product yields. Hence, a central setting (16 MPa; 40 °C) was selected to test the effect of the concentration of methanol in the solvent mixture. The results are shown in Figure 6. The concentration of benzylammonium mandelate was approximately 2.3 mg/mL in all the experiments.

The effect of the methanol concentration did not seem to be dramatic either. However, it was clearly visible that an increase in the amount of organic solvent in the high-pressure vessel affected the amount of crystalline product and extract. The tendency was as expected: the more methanol added, the larger the solubility of the racemic salt and the lower the raffinate yield.

During the extraction phase of the antisolvent fractionation experiments, the dissolving power of the solvent mixture gradually decreased. This could also mean that the boundary line of the biphasic area in the ternary solubility diagram (maroon line in Figure 1) could shift towards the solvent node. With the eutonic composition remaining in the close vicinity of its original value, the first zone of the ternary diagram could become narrower.

## 3. Materials and Methods

### 3.1. Materials

Racemic as well as enantiomeric mandelic acid and benzylamine were purchased from Sigma-Aldrich. According to the manufacturer’s specifications, their purity exceeded 99%.

Methanol was used as the solvent for all crystallization experiments as well as the solvent in the trap during the high-pressure antisolvent fractionation experiments. It was bought from two companies; Molar Chemicals and Merck. The products of both suppliers were over 99.9% pure.

The carbon dioxide (Biogon C) needed for the antisolvent fractionation experiments was ordered from Linde Gas Hungary, with over 95% purity. It was used freshly distilled in all the experiments.

### 3.2. Atmospheric Recrystallization

The ambient pressure partial recrystallization experiments were carried out in simple laboratory vials. Enantiomeric mixtures of benzylammonium mandelate were dissolved in methanol at the desired concentrations. Dissolution was aided by an ultrasonic bath at room temperature. The crystallization of the solute was induced by cooling the vials to 1 °C in a refrigerator, where they were left to equilibrate for a minimum of 3 days. A sample was then taken from the mother liquor of each vial, which was then diluted and stored for the capillary electrophoresis analyses. Vacuum filtration (on a glass filter with a G1 class meaning 90–150 µm pore diameter) was used to separate the crystals from the remaining mother liquor. In the preliminary dissolution tests, the solubility of the salt in acetone was found to be little. Hence, chilled acetone was used to rinse the crystal from the traces of the mother liquor. The vacuum was only stopped when the solid appeared to be dry. A fraction of the crystals was dissolved in methanol at an appropriate concentration for capillary electrophoresis. The course of the capillary electrophoretic measurements was identical to those described in our earlier studies [12,22,24].

### 3.3. Gas Antisolvent Fractionation

The GASF experiments were conducted in a laboratory autoclave with approximately a 37 mL internal volume. The vessel was equipped with pressure and temperature transducers and its contents could be mixed using a magnetic mixer and rod. The schematics of the autoclave can be found in earlier communications. Mandelic acid of the desired initial enantiomeric purity was mixed with an equimolar quantity of benzylamine and dissolved in methanol. The total mass of the salt was set based on its concentration to be achieved in the autoclave. A total of 4 mL of the organic solution was then loaded into the high-pressure vessel. The quantity of each component and the loaded solution were confirmed by weighing. The autoclave was then sealed and pressurized CO_2_ was introduced from a Teledyne ISCO 260D syringe pump. The pump was operated in a constant pressure mode and was tempered. The volume change in the pump was registered and later used to calculate the mass of CO_2_ inside the autoclave based on the density of carbon dioxide. The database of NIST was used to obtain the density values for the pressure and temperature of the pump’s cylinder. After loading all the components, a roughly one hour equilibration phase followed with the mixer set to 1000 rpm. In order to obtain a (seemingly) dry product and to carry out the separation of the precipitated and dissolved fractions, the reactor was rinsed with pure CO_2_. The quantity of CO_2_ was pre-determined to cover approximately three times the volume of the autoclave. The exiting gaseous stream was bubbled through a solvent trap (a flask filled with methanol). Afterwards, the vessel was depressurized. The outlet pipe was washed with 5 mL of methanol into the solvent trap, which was referred to as the extract. The solid product (the raffinate) was collected directly from the autoclave. The remainder was washed out with methanol. The solvents from the trap and the dissolved raffinate were evaporated and the mass balance was established.

### 3.4. Experimental Investigation of the Chiral Melting Phase Diagram

As discussed in the introduction, the chiral melting phase equilibrium diagram carries valuable information about the expectable rate of enantiomeric enrichment. To predict the liquidus curve on the diagram, the melting temperature of the racemate and the pure enantiomer as well as their fusion enthalpies were determined. Other measurements were conducted at arbitrary enantiomeric excess values to confirm the prediction results.

A TA instrument DSC cell 2920 heat flux-type differential scanning calorimeter was used in the measurements. The samples were prepared by the dissolution of the scalemic acid and the equimolar base in methanol and then the solvent was evaporated. Roughly 2 mg of the carefully powdered crystalline material was filled into an aluminum pan, which was then sealed. Heating cycles were conducted without a gas purge in unpunctured sample holders. A 5 °C/min heating ramp was applied from room temperature to 130 °C, where a 3 minute plateau was inserted. It was necessary to equilibrate the samples before changing to a slower heating rate of 2 °C/min up to 230 °C through the expected range of the ambient melting temperature of the salt. Eutectic melting was evaluated by the peak maximum and the non-eutectic melting temperatures were considered to be the onset temperatures of the corresponding peak.

In the approximative prediction of the melting liquidus curve, the Schröder–van Laar and Prigogine–Defay equations were used as they are commonly applied in the calculation of melting phase equilibria.

### 3.5. Evaluation of the Recrystallization Experiments

In all the recrystallization experiments, regardless of being atmospheric or high-pressure, the yields of the crystalline phase (raffinate) and mother liquor (extract) ($Y_{i}$) were identically evaluated based on the masses of the dried product ($m_{i}$), with *i* being substituted for the product and the mass of the salt dissolved in the organic solvent ($m_{0}$). In the atmospheric experiments, only the mass of the crystalline product could be accurately measured; hence, that of the salt from the extract was calculated assuming a total compound recovery.
(1)$$Y_{i}=\frac{m_{i}}{m_{0}}$$

The chiral composition of the products was characterized using chiral capillary electrophoresis. The enantiomeric excess values ($ee_{i}$) were calculated using the electrophoretic peak areas for the *R* and *S* enantiomers $A_{R}$ and $A_{S}$, respectively.
(2)$$ee_{i}=\left|\frac{A_{R}-A_{S}}{A_{R}+A_{S}}\right|$$

The enantiomeric excess of the initial mixtures ($ee_{0}$) was calculated based on the mass of the racemic ($m_{rac}$) and enantiomeric ($m_{ena}$) mandelic acid that was weighed into the glass vials we used to prepare the starting organic solutions.
(3)$$ee_{0}=\frac{m_{ena}}{m_{rac}+m_{ena}}$$

In the atmospheric recrystallization experiments, the triangular solubility diagram of the enantiomeric system, benzylammonium mandelate, and the solvent, methanol, was constructed. The compositions in the mother liquors of the experiments were used as the crystallizing vials were always left for multiple days to equilibrate. 

## 4. Conclusions

Chiral melting phase diagrams are often mirrored by ternary solubility diagrams. In thermodynamically controlled recrystallization processes, such ternary solubility diagrams affect the products of enantiomeric enrichment processes. The salt of mandelic acid and benzylamine was used as an example for the experimental determination of a ternary solubility diagram.

Gas antisolvent fractionation—an alternative, innovative crystallization process—was used to exploit the self-disproportionation of enantiomers in the enantiomeric enrichment of the same salt. The traces of the influence of the ternary solubility diagram could be discovered among the results. However, the connection was not as direct as in the case of atmospheric recrystallization procedures, probably because of the changing composition of the solvent mixture during the extraction phase.

## Figures and Tables

**Figure 1 molecules-28-02115-f001:**
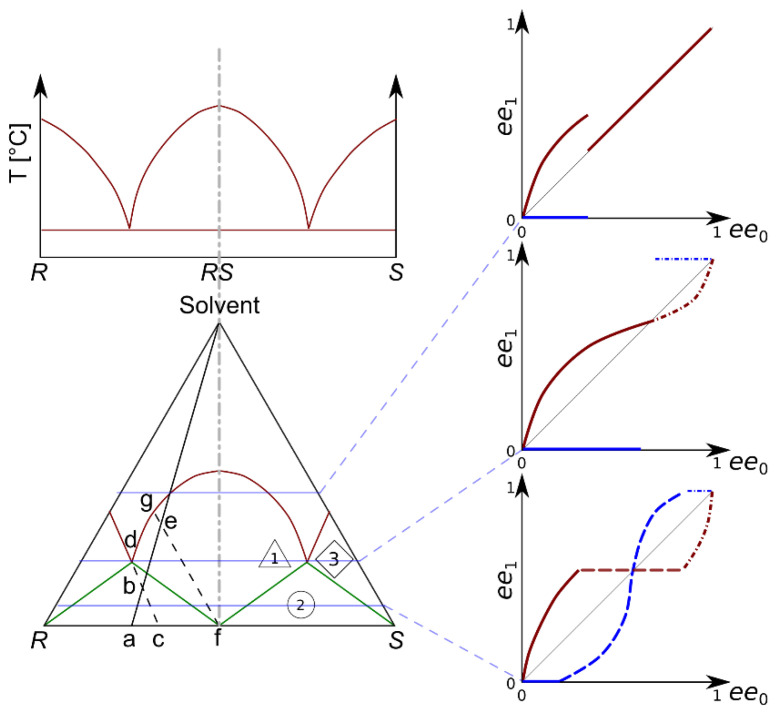
Comparison between a chiral temperature composition melting phase diagram and the recrystallization behavior of a hypothetical racemic compound. The diagrams on the left show the correlation between the melting phase equilibrium and the zones of the ternary solubility plot, whilst the *Product ee* against *Initial ee* diagrams on the right show the expected (idealized) product compositions in the case of different overall concentrations. The expected composition of the mother liquor is shown in maroon and that of the crystalline product is shown in blue.

**Figure 2 molecules-28-02115-f002:**
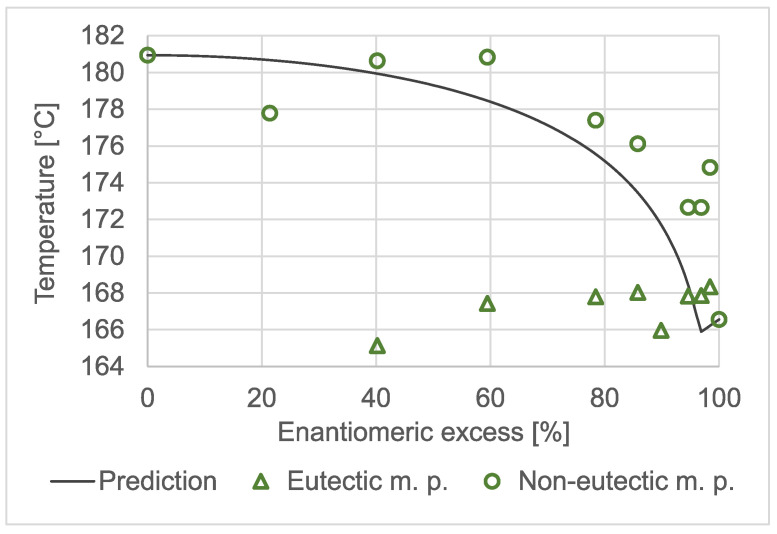
The chiral temperature composition phase diagram of the equimolar salt of scalemic mandelic acid and benzylamine.

**Figure 3 molecules-28-02115-f003:**
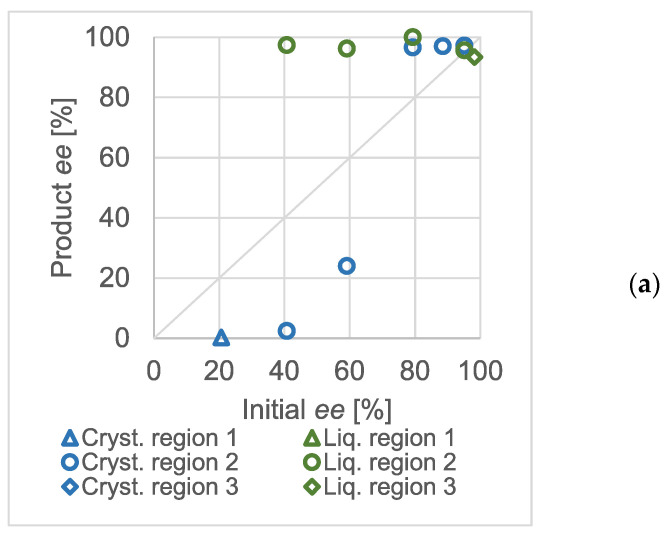
Enantiomeric enrichment data of scalemic, equimolar benzylammonium mandelate salts at an ambient pressure at 1 °C. (**a**) Total salt concentration: 124.9 mg/mL; (**b**) total salt concentration: 85.1 mg/mL; (**c**) total salt concentration: 58.5 mg/mL; (**d**) total salt concentration: 29.38 mg/mL.

**Figure 4 molecules-28-02115-f004:**
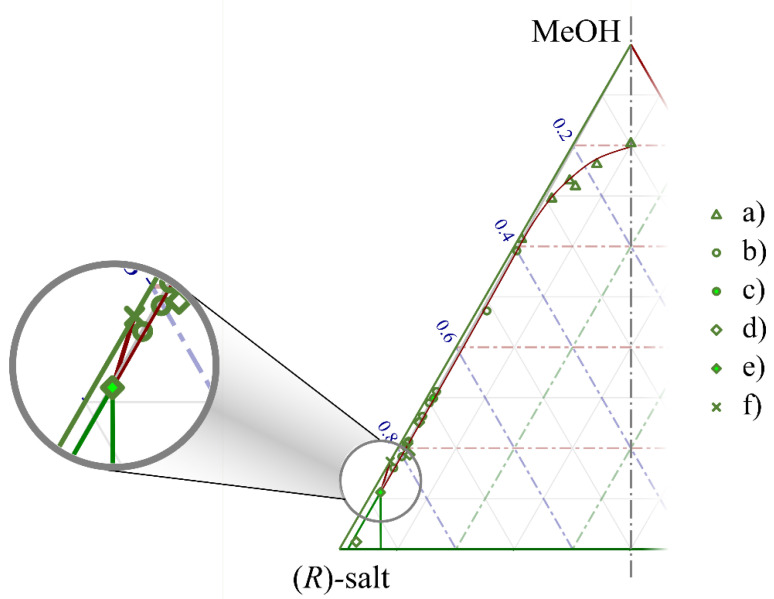
The estimated ternary solubility plot (shown along its axis of symmetry) of benzylammonium mandelate at ambient pressure and 1 °C using methanol as a solvent. (**a**) Nearly racemic crystals and scalemic mother liquor (boundary of zone 1); (**b**) scalemic crystals and eutonic mother liquor (boundary of zone 2); (**c**) average of the points in dataset ‘b’; (**d**) nearly enantiomeric crystals and scalemic mother liquor (boundary of zone 3); (**e**) average of the data in ‘d’; (**f**) solubility of the pure enantiomer (the scaling on the side of the diagram corresponds with 100 times the mole fraction of the given species).

**Figure 5 molecules-28-02115-f005:**
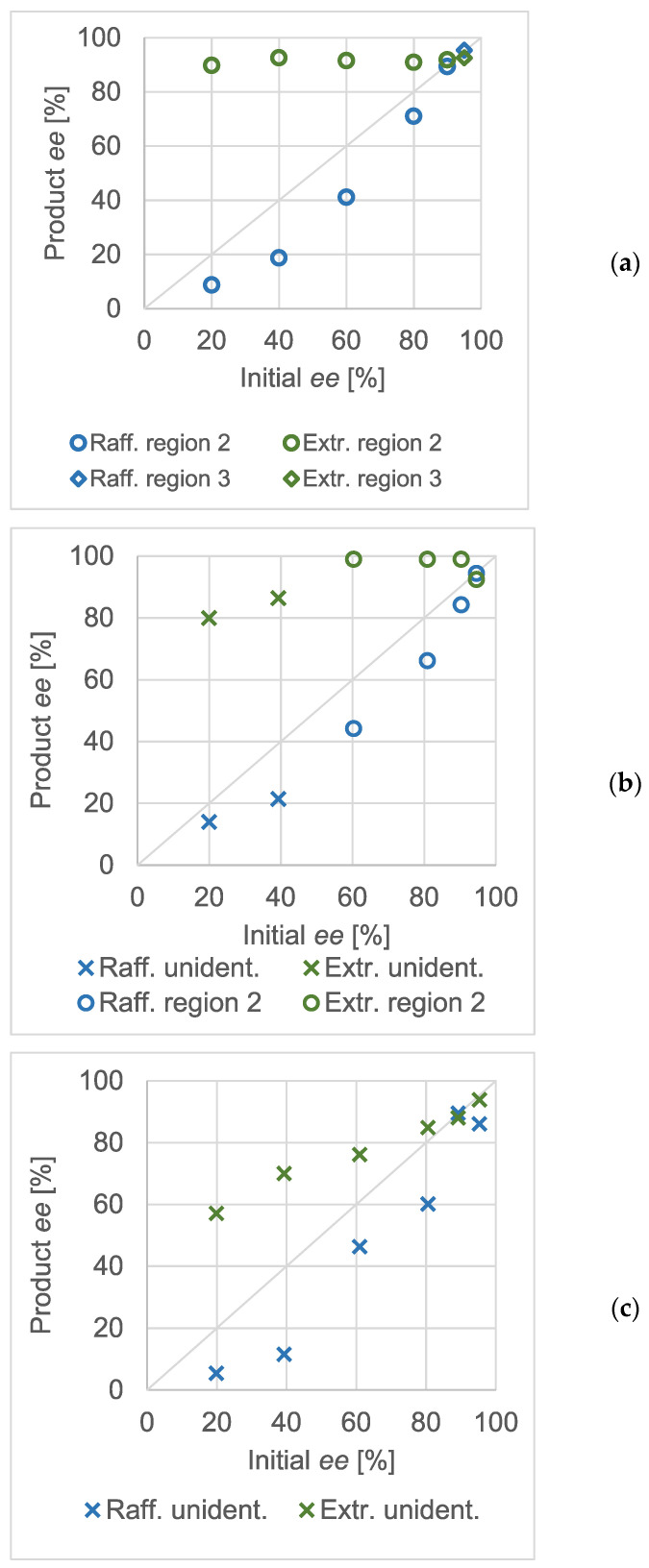
Enantiomeric enrichment via gas antisolvent fractionation at 20 MPa and 35 °C. The concentration of methanol was constant; approximately 90 mg/mL. (**a**) The overall concentration of the salt was 2.3 mg/mL. (**b**) The overall concentration of the salt was 1.15 mg/mL. (**c**) The overall concentration of the salt was 0.60 mg/mL.

**Figure 6 molecules-28-02115-f006:**
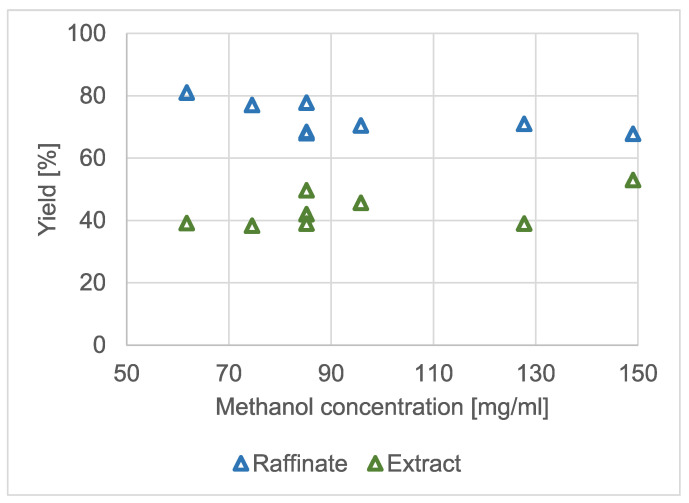
The effect of the concentration of methanol in the autoclave during the recrystallization of racemic benzylammonium mandelate.

## Data Availability

The data presented in this study are available in the manuscript or upon request from the authors.

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
