# Peer review of "Experimental Determination of a Chiral Ternary Solubility Diagram and Its Interpretation in Gas Antisolvent Fractionation"

_molecules, 2023, doi:10.3390/molecules28052115_

Round 1

Reviewer 1 Report

Title: OK

Abstract:  OK

Introduction: In general, the introduction is well written, the topic is well contextualized, the knowledge gap is clearly presented, and finally, the objective of the research is presented.  However, citation should be improved, as many of the ideas and information reported lack academic support. Example: Improve paragraph citations between lines 27-35 (no citations). 

Results and discussion:  1) Some of the results such as the SDC analysis have not been discussed. 2) Define clearly throughout the document, which is the salt referred to in the analysis ... example: Figure 3 "concentration of the salt is 2.3 mg/m" which salt? 

Materials and Methods: 1) Renumber, change 4 to 3 2) In section 4.3 (renumber to 3.3) the DSC determination is reported, however there is no analysis of the information in the "results and discussion" section. 3) Describe in detail the method of quantification of each of the chemical species. It is not clear how the racemic concentration was determined experimentally. 

Author Response

We would like to thank the Reviewer for reading and evaluating our manuscript submitted for publication in the Molecules journal. In the following paragraph, we are answering the comments of the Reviewer from chapter to chapter, as they have listed the issues found in the manuscript.

Introduction:

In order to support the statements on the history and current state of chirality-related research, we have included new references in the paragraph. However, we believe that the further parts of the Introduction are written with appropriate citations as the cited references cover more generalised works discussing the principles of chiral separations rather than actual examples.

Results and discussion:

The chiral melting phase-diagram of the benzylammonium – mandelate salt has been presented in the original manuscript as Figure 2, and discussion was also written in chapter 2.1. We have extended the chapter by providing the measured melting temperatures and fusion enthalpies of the racemate and the enantiomer. These values were used in the predictive calculation of the liquidus curve marked with the solid line.

We have inserted an explanation of how we use the word ‘salt’ at the end of the Introduction chapter. However, as only the equimolar salt of mandelic acid and benzylamine was used in the study, the captions of the figures were left unchanged to retain a possibly shorter length.

Materials and methods:

Thank you for your comment about the section numbers; the numbering was faulty and has since been corrected.

The results of the DSC analyses have been discussed in section 2.1. We have mainly used the DSC analyses to predict and verify the liquidus curve of the benzylammonium mandelate salt. Using this information, the racemic nature of the chemical was confirmed. The range of the eutectic composition has also been determined. In thermodynamically controlled partial crystallisations, it correlates to the expectable limiting compositions in fractioned recrystallisations.

The capillary electrophoretic measurements are not discussed in detail in the current manuscript. The method has been discussed in our earlier work and has been used without any changes in the current study. However, based on the Reviewer’s comment, we have improved the citation of the articles that contain the accurate description of these analyses.

Reviewer 2 Report

I have read the report by Székely et al. which continues on with their work on GASF. The work is quite straightforward and attempts with some success to predict the GASF of mandelic acid–benyzlamine. I think it is a worthwhile contribution to the area and chiral studies in general and recommend publication of the work. Only two minor things, on line 70, page 2, it should be region number 2, not 1, and there is no figure 2.

Author Response

We would like to thank the Reviewer for their consideration of our manuscript. We have proofread the text again and found numerous grammatical mistakes, which have been corrected. We paid particular attention to the issues mentioned by the Reviewer: the mentioned paragraph on page 2, around line 70 has been rephrased. Figure 2. and its discussion are now present in the manuscript.

Round 2

Reviewer 1 Report

The authors made the suggested changes